# Aliquots of MIL-140 and Graphene in Smart PNIPAM Mixed Hydrogels: A Nanoenvironment for a More Eco-Friendly Treatment of NaCl and Humic Acid Mixtures by Membrane Distillation

**DOI:** 10.3390/membranes13040437

**Published:** 2023-04-17

**Authors:** Giuseppe Di Luca, Guining Chen, Wanqin Jin, Annarosa Gugliuzza

**Affiliations:** 1Institute on Membrane Technology, National Research Council (CNR-ITM), Via Pietro Bucci 17C, 87036 Rende, Italy; g.diluca@itm.cnr.it; 2State Key Laboratory of Materials-Oriented Chemical Engineering, College of Chemical Engineering, Nanjing Tech University, 30 Puzhu Road, Nanjing 211816, China; gnchen@njtech.edu.cn (G.C.); wqjin@njtech.edu.cn (W.J.)

**Keywords:** layer-by-layer, thermal and pH responsive membranes, responsive interfaces, water purification, membrane distillation

## Abstract

The problem of water scarcity is already serious and risks becoming dramatic in terms of human health as well as environmental safety. Recovery of freshwater by means of eco-friendly technologies is an urgent matter. Membrane distillation (MD) is an accredited green operation for water purification, but a viable and sustainable solution to the problem needs to be concerned with every step of the process, including managed amounts of materials, membrane fabrication procedures, and cleaning practices. Once it is established that MD technology is sustainable, a good strategy would also be concerned with the choice of managing low amounts of functional materials for membrane manufacturing. These materials are to be rearranged in interfaces so as to generate nanoenvironments wherein local events, conceived to be crucial for the success and sustainability of the separation, can take place without endangering the ecosystem. In this work, discrete and random supramolecular complexes based on smart poly(N-isopropyl acrylamide) (PNIPAM) mixed hydrogels with aliquots of ZrO(O_2_C-C_10_H_6_-CO_2_) (MIL-140) and graphene have been produced on a polyvinylidene fluoride (PVDF) sublayer and have been proven to enhance the performance of PVDF membranes for MD operations. Two-dimensional materials have been adhered to the membrane surface through combined wet solvent (WS) and layer-by-layer (LbL) spray deposition without requiring further subnanometer-scale size adjustment. The creation of a dual responsive nanoenvironment has enabled the cooperative events needed for water purification. According to the MD’s rules, a permanent hydrophobic state of the hydrogels together with a great ability of 2D materials to assist water vapor diffusion through the membranes has been targeted. The chance to switch the density of charge at the membrane–aqueous solution interface has further allowed for the choice of greener and more efficient self-cleaning procedures with a full recovery of the permeation properties of the engineered membranes. The experimental evidence of this work confirms the suitability of the proposed approach to obtain distinct effects on a future production of reusable water from hypersaline streams under somewhat soft working conditions and in full respect to environmental sustainability.

## 1. Introduction

Water pollution, extraction from groundwater basins, climate changes, and geochemical cycles are some of the major causes of risk for human health, biodiversity, and planet survival [1,2]. Water stresses as well as upward domestic, industrial, and livestock water consumption are seriously compromising the quality and quantity of usable water [3,4,5]. For such a reason, there is vigilant activity in identifying competitive and eco-sustainable technologies that support sophisticated and sustainable water management practices [6,7,8]. Membrane distillation (MD) is an example of green-powered technology, which allows freshwater to be recovered from wastewater and seawater through changes in the state of water [9,10]. In the most common thermally driven direct contact (DC) configuration, the MD technology uses hydrophobic porous membranes to separate two aqueous phases, one containing the stream to treat and the other containing pure water [11]. Applying a difference of temperature across the membrane, the water is evaporated from the hot side (feed = aqueous stream), is diffused as a vapor through the pores of the membrane and, is then collected at the cold side after condensation (permeate = pure water). A suitable combination of chemistry and morphology can make the membrane waterproof and permeable to water vapor at the same time [12,13]. In the logic of energy sustainability, MD can be also powered by solar, wind, and wave energy [14,15,16]. The competitiveness and sustainability of this technology is, however, strongly dependent on the choice of the membrane.

A membrane with hydrophobic properties is strongly desirable. Polymers such as polypropylene (PP), polytetrafluoroethylene (PTFE), and polyvinylidene fluoride (PVDF) are in fact the most used. PTFE is a highly crystalline polymer with excellent thermal stability and chemical resistance. However, the sintering method combined with the melt extrusion method is the unique practical route for the fabrication of hydrophobic membranes. PP is another crystalline polymer used to fabricate porous membranes by stretching and melt extrusion processes. However, PP membranes exhibit higher surface free tension and moderate thermal stability at high temperature. Unlike PTFE and PP, PVDF is marked by better solubility in common organic solvents and is easily adaptable when being moulded in distinctive and modifiable morphologies through the use of different and combined manufacturing techniques, including well-established phase separation. It is also a thermally stable and highly hydrophobic polymer with high resistance to most corrosive chemicals and organic materials. Over the last few years, PTFE, PP, and PVDF polymers and related copolymers have been used to fabricate commercial hydrophobic membranes, which have been initially applied in microfiltration and successively adapted to MD applications. While PTFE and PP membranes are often used in commercial and pilot MD systems, the potential of PVDF membranes in making MD operations much more competitive at scale is still under investigation. However, attempts at solutions in that direction have provided reasonable and encouraging results in the recent past [17]. Despite the fact that MD processes based on PVDF membranes are still a technology validated in a relevant environment, a lot of research has been carried out to explore the potential of this polymer in combination with other hydrophilic, organic, and inorganic materials. This is because improved surface and transport membrane properties are highly desirable for scaled MD processes [17]. For handling, PVDF is indeed one of the most used polymers for the fabrication of high-performing nanocomposite hydrophobic membranes, which are expected to catalyze the passage from traditional physical barriers to interactive chemical interfaces with amplified performance [18,19,20,21,22,23,24,25,26,27,28,29].

Among the various materials used, 2D materials have been demonstrated to be particularly attractive for improving the outputs of membrane separations [23,26,28]. However, less consideration has been given to managing their quantities with a regard for safety and respect for the environment. A desired target is to use the minimum quantity of nanofiller to enhance water yield. An option could be to confine small quantities of materials to the membrane surface considering that interfacial interactions between the membrane and surrounding environment can decide the final result [23,30,31,32,33,34,35]. In every membrane process, including MD operations, engineered membrane surfaces are in fact already expected to establish selective interactions with approaching solutions at the early stage [36,37,38,39,40,41,42,43,44,45,46,47,48,49]. Affinity and repulsive forces can be established at the interface so that permeation and/or refoulement can be well addressed.

This study provides experimental evidence about the effects of a dual responsive nanoenvironment, generated throughout the membrane surface, on the sustainability of the overall MD process. Aliquots of nanofiller are adhered to the membrane skin as a practical route for promoting sub-nanometer control over the surface properties so as to obtain (a) enhanced production of freshwater from hypersaline solutions containing NaCl and humic acid, (b) contrasted adhesion of foulants on the membrane surface, and (c) in situ self-cleaning action without the use of additional harsh chemical agents.

Specifically, a layer-by-layer (LbL) spray technique [50,51,52,53,54,55,56,57,58,59,60] is used for adsorbing randomly discrete complexes of a Zr-MOF compound and graphene and hydrogels on the surface of a PVDF membrane. While graphene together with other 2D materials has been successfully tested in MD operations [23,26,35,61,62,63,64,65,66,67,68,69,70,71], there is not yet a solid indication about the potential of metal–organic framework compounds (MOFs) in DCMD applications [72,73,74].

Large specific surface area, regular porous structure, and good thermal and chemical stability have made MOFs of significant interest for gas separation, catalysis, storage, and drug delivery [75,76,77,78,79], while their effective role in MD has yet to be proven. Cao et al. [67] proposed a study of molecular dynamic simulation through an ultrathin conductive MOF film, revealing the capability of the latter to permeate water to be three to six orders of magnitude higher than traditional membranes and one order of magnitude higher than single-layer nanoporous graphene or molybdenum disulfide (MoS_2_). Yang et al. [72] proposed the fabrication via electrospinning of superhydrophobic poly(vinylidene fluoride) with Fe-MOF up to 5 wt.% for equipping DCMD devices. Fluxes up to 2.87 Lm^−2^h^−1^ have been obtained along with NaCl rejection of 99.99%. Cheng et al. [74] operated aluminum fumarate MOF/PVDF hollow fiber membranes in DCMD plants, yielding fluxes of 8.04 Lm^−2^h^−1^ at 50 °C and 15.64 Lm^−2^h^−1^ at 60 °C. A gain in flux was obtained by MOF-functionalized alumina tubes with values of 16.7 Lm^−2^h^−1^ at 50 °C and 32.3 Lm^−2^h^−1^ at 60 °C, though working in a more expensive vacuum configuration (VMD) [80]. In all these cases, MOFs have been used as physical spacers inside the polymer matrix to increase the intrinsic porosity of the membrane and open additional free gaps among the polymer chains. In this way, resistance to mass transfer was reduced and water flux was increased.

In the present work, we confine aliquots of ZrO(O_2_C-C_10_H_6_-CO_2_) (MIL-140) and graphene to the surface of a PVDF membrane with the intent of triggering effective chemical interactions without affecting the original morphology and packing of the host polymer matrix. We use complexes of ionic thermo- and pH-responsive poly(N-isopropyl acrylamide) (PNIPAM) mixed hydrogels, i.e., acid- and amine-terminated hydrogels, to provide 2D materials with a chemical environment allowing (a) a random and electrostatically driven deposition of the nanofillers over the membrane skin (active surface), (b) hydrophobic thermal effects that prevent detachment or leaking of the adhered materials in aqueous media, and (c) modulated negative charge density for improved resistance to fouling and the stimulation of self-cleaning actions. 

Random and discrete aliquots of MIL-140 and graphene have proven to be effective in enhancing water flux through cooperative interfacial forces established at the membrane–solution interface. The ‘hydrogel–2D materials’ complexes have also been demonstrated to contrast fouling events and remove foulants away from the surface through a simple switch of the ionic charge. Repulsion forces are further addressed at facilitating eco-friendly cleaning procedures, with a full recovery of the initial performance of the membranes bringing charged surfaces. These types of engineered membranes are promising and seem to bridge the distance between more traditional and new attractive families of responsive interfaces, which in the near future could make DCMD a more reliable, efficient, and environmentally friendly operation for the recovery of freshwater.

## 2. Experimental Section

### 2.1. Materials

PVDF (Solef^®^6020, Solvay Solexis: water adsorption <0.040% at 23 °C after 24 h; dp = 1.78 kg/m^3^) was kindly supplied by Solvay Specialty Polymers (Milan, Italy). Graphene (G) flakes were purchased from Sigma Aldrich (carbon > 95 wt.%; oxygen < 2 wt.%, Milan, Italy). MIL-140 [ZrO(O_2_C-C_10_H_6_-CO_2_)] (*M*) was synthesized according to the procedure reported in [81]. This choice was due to the fact that Zr-based MOFs are among the most water and solvent stable and mechanically resistant due to the strong Zr-O bonds [68]. N-Methyl-2-pyrrolidinone (NMP, Riedel de Häem: max 0.05% in water, d. 1.03 kg m^−3^) and 2-propanol (IPA, WWR PROLAB: d. 0.78 kg m^−3^, Milan, Italy) were used as the respective solvent and nonsolvent for the preparation of microporous PVDF membranes. N-ethyl-o,p-toluenesulphonamide (Sigma Aldrich, water solubility < 0.01 g/100 mL at 18 °C, Milan, Italy) was used as a WS. Carboxylic acid-terminated poly(N-isopropylacrylamide) (A, M_n_ 5000) and amine-terminated poly(N-isopropylacrylamide) (N, M_n_ 5000) were purchased from Sigma Aldrich. Fluorinert (FC-40, Novec, Merck, Milan, Italy) was used for gas–liquid displacement measurements for pore size and overall porosity estimation. Ultra-pure water (filtered by a USF ELGA plant) was used to investigate membrane anti-wetting properties. NaCl with a degree of purity of 100% was purchased from VWR Chemicals. Humic acid (HA, Sigma Aldrich, Milan, Italy) was mixed with NaCl for hypersaline solutions. All materials were used as received. 

### 2.2. Membrane Preparation

PVDF powder (12 wt.%) was dissolved and stirred for 24 at 40 °C in NMP. The homogeneous solution was cast on a glass support through the use of a micrometric film applicator (Elcometer) with a gap size of 250 μm. The casting solution was then coagulated in 2-propanol and washed with deionized water. After drying at room temperature, the membrane was further treated at 40 °C for 1 h and then sprayed with a solution of N-ethyl-o/p-toluenesulphonamide dissolved in ethanol at 1.5 w%. PNIPAM-NH_2_ (N) and PNIPAM-COOH (A) were dissolved in water (10^−2^ M based on the repeat unit molecular weight) at 16 °C while MIL-140 (M) and graphene (G) were dispersed in PNIPAM-NH_2_ under ultrasound treatment for 24 h at a concentration of 1.0 mgmL^−1^. The pH value of each solution was adjusted to 5.7, which is typical of solutions containing humic acid.

A PVDF membrane was used as the substrate for discrete and random deposition of ionic PNIPAM/nanofillers complexes. The solutions were alternatively sprayed five times starting from the amine-terminated hydrogel (N) using a commercial glass spray bottle held 50 cm from the sublayer; a waiting period of 10 min was allowed between each layer to facilitate adsorption. For simplicity, these hybrid complexes were named as LAN in absence of 2D materials, LANG and LANM with graphene and Zr-MOF, respectively, and LANMG with graphene alternated with Zr-MOF (Figure 1). All samples were dried in the air before testing.

### 2.3. Methods

Membrane morphology and topography features were examined using Scanning Electronic Microscopy (SEM, Zeiss EVO MA10, Oberkochen, Germany) and Atomic Force Microscopy (AFM, Nanoscope III Digital Instruments, VEECO Metrology Group, Santa Barbara, CA, USA). The latter was operated in tapping mode at a rate of 1 Hz across sample surfaces for 512 points. The pore size of the membranes was measured by gas–liquid displacement using a porosimeter (Capillary Flow Porometer-CFP 1500 AXEL, Porous Materials Inc., Ithaca, NY, USA). Overall porosity was measured by filling the membranes with FC-40. Membrane weight was estimated before and after filling and porosity was expressed in a percentage as the ratio between the volume occupied by the fluorine liquid and the volume of the membrane. The amount of materials deposited was estimated after drying through the use of a balance with five digits. Deposition, stability, and the thermo-responsive behavior of the ionic PNIPAM hydrogels were investigated by infrared spectroscopy in ATR mode (Spectrum One System, Perkin Elmer, Milan, Italy). MOF adsorption was also inspected by EDX (Zeiss EVO MA10, Oberkochen, Germany). Waterproofness was estimated by measuring the contact angle values throughout the functionalized surfaces (Cam200 KSV instruments, LTD, Helsinki, Finland).

### 2.4. Membrane Distillation Tests

Thermally driven DCMD experiments were executed using NaCl (35 gL^−1^) and mixtures of NaCl/HA (35 gL^−1^/1.0 mgmL^−1^), flow rates of 6 Lh^−1^ (feed side) and 4.98 Lh^−1^ (permeate side), and temperatures of T_feed_ = 45 °C and T_perm_ = 16 °C. Retentate and distillate streams were converged in a counter-current way toward the membrane module containing the membrane, where the liquid water was evaporated. On the retentate side, a pump was taking and sending the heated feed to the membrane module. Additionally, on the distillate side, a second pump ensured the counter-current recycling of the cold stream in order to remove the vapor diffusing through the pores of the membrane from the solution. The trans-membrane fluxes were calculated by weighing the variations in the distillate tanks. The experiments were run from 6 to 18 h continuously. The salt conductivity of the feed and permeate streams was measured at the end of every single experiment through the use of a conductive meter (HI 2300 bench meter supplied by Hanna Instruments, Woonsocket, RI, USA).

## 3. Results and Discussion

### 3.1. Membrane Fabrication

On the assumption that each membrane is crucial for molecular separation, new concept membranes are here proposed to make water purification more eco-friendly when a DCMD operation is performed. Engineered membrane surfaces have been designed through combined phase inversion, WS, and LbL spray procedures. The intent is to provide a greener route for the fabrication of new functional membranes designed for more sustainable management of water.

Figure 1 displays the surface and a cross section of PVDF membranes before and after surface engineering. It is pertinent to consider the spherulitic-like structure of the PVDF membrane induced by the exchange of solvents. This kind of morphology is due to crystallization events, which take place during phase inversion. Because PVDF is a semi-crystalline polymer, well-sized spherulites interlinked by polymeric filaments are formed during the delayed solid–liquid demixing of the polymer solution, as displayed in the image included in Figure 1a’. Free gaps are generated between the polymeric particles through the overall symmetrically structured film and work as effective pores of the membranes (Table 1). This singular topography is also somewhat attractive for its high resistance to wetting (*θ* = 130 ± 2°), which can be regarded as an effect of the high irregularity of the surface (*Rq* = 112 nm). Highly interconnected open paths together with high waterproofness make this kind of PVDF membrane a suitable candidate for MD applications.

To make this PVDF membrane a well-suited sublayer for PNIPAM hydrogels, the film was subjected to WS treatment through the use of a solution of N-ethyl-o/p-toluenesulphonamide in ethanol. This organic compound has typical amphiphilic properties and very low water solubility, with subsequent good affinity towards materials with different chemistry and resistance to aqueous media. Separately, colloidal dispersions of MIL-140 and graphene were prepared in the hydrogel-NH_2_. The pH value of all solutions was adjusted to 5.7 so that the density of charge generated through the hydrogel segment chains allowed ion pairs to be formed by electrostatic attraction. It is relevant to observe how no continuous nano-films, but rather discrete aggregations of the complexes, were randomly deposited through the surfaces without affecting the bulk of the membrane (Figure 1b,b’). SEM images reveal the formation of fibrotic cords without occlusion of surface pores or penetration inside the membrane structure. All porosity, pore size, and pore distribution values calculated for the engineered PVDF membranes almost overlap with those measured for pristine PVDF so as not to offer further resistance to mass transfer (Figure 2a,a’). Indeed, all membranes exhibit two clusters of pores with a mean pore size of around 0.40–0.47 μm and a smallest pore size of around 0.19 μm, while the bubble point is between 0.85 and 0.9 μm (Table 1).

However, it is important to observe that the pores around 0.4 μm in size provide the major contribution to the flow measured through the membrane. The incremental filter flow (incr. %FF) reaches the maximum value at 0.4 μm (Figure 2b,b’), providing an indication of the predominance of this pore cluster.

A morphological comparative analysis between PVDF membranes before and after functionalization (Figure 2 shows a comparison between pristine PVDF and PVDF-LANM membranes) reveals that the intrinsic structural features of the membranes do not undergo substantial changes. All of this is convenient if we consider that high interfacial area, which is crucial to water vapor diffusion during MD operations, is preserved.

### 3.2. Characterization of LAN Complexes

The formation, chemical stability, and thermal responsiveness of polycation/polyanion (LAN) complexes were investigated through infrared analysis (Figure 3). Firstly, ATR spectra collected onto neat hydrogels yielded a clear indication of the typical frequencies associated to the amide group, with band I located at 1639 cm^−1^ and band II positioned at 1538 cm^−1^, while C-N stretching was detected at 1458 cm^−1^ (Figure 2a). For PNIPAM-COOH, an additional very weak absorption associated with carboxylic acid is detected at 1710 cm^−1^, while a broad absorbance between 3600 and 3200 with a maximum intensity at 3291 cm^−1^ is ascribed to the overlapping of O-H and N-H stretching modes. 

Figure 3b shows how changes in the carbonyl amide vibrations—*ν_s_* 1647 cm^−1^ and 1542 cm^−1^—take place as the LAN complex is deposited onto the PVDF surface. Shifted carbonyl frequencies as well as broad flattening of O-H and N-H vibrations can be detected. The asset of the carbonyl bands is further modified when Zr-MOF is deposited on the surface of PVDF membranes through the use of ionic hydrogels (Figure 3c,d). In this case, an intensification of the carbonyl bands along with the appearance of various shoulders and a new peak around 1492 cm^−1^ can be appreciated. In this region of the spectrum, these overlapped and distinct new vibrations can be ascribed to the asymmetric and symmetric stretching of the carboxylate groups *ν*(COO^−^), which are part of the ZrO(O_2_C-C_10_H_6_-CO_2_) structure. Additionally, the intensity and broadness of the O-H stretching mode appears to be more intense around 3200 cm^−1^. In the case of graphene, no substantial modifications are appreciated, while the spectrum collected on the membrane surface functionalized with the complex containing alternated aliquots of graphene and Zr-MOF (LANMG) shows three carbonyl stretching patterns, one of which is located at 1738 cm^−1^ (Figure 3c,d). This band is typical of the *ν*(C=O) mode of non-coordinated and free carboxylic acid groups. It should be noted that amine-terminated hydrogel bringing Zr-MOF and graphene was alternated with acid-terminated hydrogel. This obtains amine- and acid-terminated hydrogels with greater ability to neutralize the counter charges during deposition of the practically inert graphene, thus yielding more available COOH moieties of Zr-MOF in the successive step.

EDX confirms the significant presence of Zr in the chemical composition of the MIL-140 particles adhered to the PVDF membrane surface (Figure 4), while SEM and AFM micrographs distinguish the presence of the nanofillers on the membrane surfaces (Figure 4 and Figure 5). In the case of AFM, topographical bright and edged regions are well distinct from the sublayer, also yielding an indication of the irregularity of the surface (Figure 5a). The random deposition of discrete objects produces an increase in the roughness factor, with a subsequent improvement in water repellence (Figure 5b). At equilibrium, membranes functionalized with complexes containing graphene (LANG) show a contact angle value of 136 ± 4° against the 134 ± 3° estimated for membranes containing Zr-MOF alone (LANM). For membranes with mixed Zr-MOF and graphene (LANMG), the contact angle value undergoes a negligible decrease (133 ± 3°) due to the higher availability of free carboxyl groups, while a value of 129 ± 4° was measured on membranes functionalized with ionic hydrogels without 2D materials (LAN) (Table 1). While the liquid water entry pressure (LEP_w_) is less than 1 bar for all membrane samples, the singular topography of these hybrid membranes provides good resistance to liquid spreading due to the fact that the morphological component moves the surface properties towards higher irregularity and a subsequent reduced contact line between liquid and polymer (Figure 5c). In this case, 2D materials amplify the irregularity of the section profile, leading to a good fit between the root mean square deviation (*R_q_*) and contact angle values (Table 2). Generally, increasing the value of surface roughness improved the anti-wetting behavior of engineered surfaces, thus leading to contact angle values of up to 137 ± 4°. Regarding this point, a short premise is required. MD requires operating hydrostatic pressure lower than that of pressure-driven processes such as reverse osmosis (RO). It works at pressures near that of atmospheric pressure since small differences in vapor partial pressure are enough to promote mass transfer. Thus, the high degree of waterproofness estimated for all proposed membranes is mostly expected to balance the low values of LEP, thereby resulting in a suitable resistance to liquid intrusion.

To evaluate the stability of the hierarchical materials in aqueous medium, membrane samples were submerged in water and stirred vigorously for 6 h at 40 °C. The films were then dried overnight and inspected again by infrared spectroscopy (Figure 6).

Comparison of the spectra collected from samples before and after treatment does not reveal substantial modifications to the intensity and vibration of the typical carbonyl bands, thereby suggesting the hydrophobic state of materials when operated at temperatures above 40 °C rather no leakage. It is known that PNIPAM hydrogels exhibit hydrophobic properties in aqueous solution at temperatures higher than the Low Critical Solution Temperature (LCST, around 32–33 °C) [82,83,84]. Above the LCST, a rigid water-soluble ‘ice-like structure’ is generated from the effect of intermolecular aggregations. Water molecules are released and hydrophobic groups are exposed to each other. In this state, the hydrogel exhibits typical hydrophobic behavior and insolubility in water. It is also known that functional side groups in the polymer, as well as changes in pH, and salts dissolved in water, may affect the LCST in the order of 0.1 to 5 °C [85,86,87]. As an example, values of 33.1 to 37.7 °C have been detected at pH 5 for some synthesized poly(NIPAM-co-AAD) copolymers and acid-terminated hydrogels. Our experiments give a clear indication about the stability of the hydrogels complexes in aqueous solutions at 40 °C and pH 5.7, denoting affinity towards wet PVDF sublayers rather than water.

### 3.3. Membrane Distillation Testing

Based on the previous results, MD experiments were carried out, with the composite–hybrid membranes exposed to saline streams at 45 °C. Under this temperature, the hydrogels were in a hydrophobic state and, hence, insoluble in aqueous media. Further, the pH value of the saline streams was around 5.7 and comparable to that of the LBL deposition. Thus, no changes in charge density are expected and the stability of the supramolecular complexes is preserved during MD. Experiments with a solution of NaCl at a concentration comparable to that of seawater (35 gL^−1^) were conducted for 18 running hours.

Figure 7a displays the trend of the average flux measured through all membranes with time. As compared to the pristine PVDF, enhanced flux is observed through all functional membranes. A gain up to 17 and 19% is estimated for membranes adhered with LANG and LANM, respectively. The increase is around 10% for membranes functionalized with the LANGM complex. This is not surprising if we consider that MOFs are rich in carbonyl sites wherein water molecules can be temporarily adsorbed and transported through additional ordered nanoporous pathways. Similarly, defective graphene has been envisaged to have a great ability to assist water diffusion [26,35]. As expected, a slight decrease in flux is detected after 18 running hours, even if mass transfer through LANM continues to be the most consistent. All engineered membranes exhibit a selectivity of 99.99% against the 99.95% estimated for the pristine PVDF after the first six hours. At the end of the test, values higher than 99.9% can be estimated for all graphene-engineered membranes, while pristine PVDF and PVDF-LANM exhibit a rejection value of 99.8% (Figure 7b).

Mixtures such as NaCl/HA (35 gL^−1^/1 mgmL^−1^) were also treated with the engineered PVDF membranes (Figure 8a). After the first two hours of operation, fluxes of up to 11.7 Lm^−2^h^−1^ were measured, with an increase of up to 58% with respect to the pristine PVDF. An increase of 30% continued to persist after six hours of operation. Figure 8b shows how the flux ratio is four to five times greater than that of the pristine PVDF during the first hour of the process. Successively, membranes with Zr-MOF show a flux ratio 1.5–1.2 times greater, indicating better performance (Figure 8b). It is relevant to observe that humic acid is strongly negatively charged at a pH greater than 4.7 [87]. Because the MD operation was performed under a pH of 5.7, membranes functionalized with aliquots of LAN and Zr-MOF exhibit a further negative charge due to the deprotonation of -COOH groups of the acid-terminated hydrogel and Zr-MOF structures. The same negative charge is consequently expected to produce higher repulsive electrostatic forces, thus leading to lower adhesion of the acid to the membrane surface. Initially, for the functional membranes, the removal rate of acid from the surface is higher than that on the neutral pristine membrane. As time progresses, the capability to contrast the acid adhesion on the surface continues to be more evident for membranes with a larger density of carboxylate groups (LANM, LANMG). 

Electrostatic repulsion has been also demonstrated to be crucial for cleaning. Reversible switching of the ionic charge throughout the surface has been proven to be enough to remove foulants from the functionalized surface quickly. To preserve the hydrophobic state of the supramolecular complexes, the membranes were washed with water at 46 °C, while the pH value was switched to a pH of 6.5 for 30 min. After a successive washing with pure water for a further 30 min, the pH was adjusted to 5.7 again and washing was continued for a further 30 min. To evaluate the recovery capability of the membranes, MD tests with pure water were further carried out. As displayed in Figure 9a, factors of recovery of 98–99% were obtained with LANM and LANMG, while a percentage of 95% was assessed for LANG against an estimated 92% for pristine PVDF. This means that strongly negatively charged membrane surfaces exhibit a better ability to remove adhered HA as an effect of repulsive forces, whereas the neutral membranes lose this capacity in the absence of additional chemical reagents. This result has to be regarded as another wide-ranging aspect of sustainability for MD processes because no additional harsh chemical agents are required to restore the quality of the engineered membranes after cleaning. Additionally, the stability and integrity of the supramolecular structures was assessed after cleaning and water permeation. Figure 9b shows no changes in intensity and carbonyl frequencies, with a more pronounced contribution of the band associated with protonated carboxylic groups observed instead due to the slight effect of the pH value. In this way, the stability and chemical steadiness of the complexes deposited on PVDF surfaces are fully satisfactory.

In summary, we demonstrated how an ionic hydrogel-based nanoenvironment (0.5 mg/cm^2^), where aliquots of 2D materials are confined, yields proficient cooperative mechanisms at the membrane surface–solution interface. Interfacial forces control water diffusion through the membranes, as comparative analyses confirm (Table 3). A practice currently in wide use is the filling of electrospun PVDF membranes with MOFs at concentrations of 0.1 to 5% in order to increase permeability under hard fluid dynamics conditions [72,73,74,88,89,90,91]. Herein, aliquots of hybrid responsive MOF supramolecular structures are demonstrated to already produce positive effects on flux under much softer operating conditions. Concerning graphene, many studies have also focused on the incorporation of few-layer, nanoplatelet, nanosheet, and quantum dot graphene in membranes [26,28,61,92,93]; however, only a few attempts have been made to adhere this material on the membrane surface for DCMD applications [35,92]. Herein, we demonstrate how the deposition of random and discrete aliquots of graphene on PVDF membrane surfaces is already beneficial to consistent productivity under mild conditions without the necessity of forming extensive coatings through more expensive procedures.

## 4. Conclusions

An eco-friendly strategy was proposed to realize new functional responsive interfaces that make membrane distillation processes much more sustainable. Zr-MOF and graphene were entangled with double responsive PNIPAM hydrogels, with discrete complexes randomly deposited on a PVDF sublayer via an LBL spray procedure. The stability and reliability of the adhered hybrid–composite structures were tuned, while the thermal and pH responsiveness of the PNIPAM was exploited to provide a hydrophobic state and a suitable repulsive environment during MD operations. The hydrophobicity of the surfaces was preserved when working at temperatures higher than the LCST traditionally observed for PNIPAM, while negative charge distribution through the membrane surface limited fouling due to HA. The hydrogels’ pH responsiveness further addressed cleaning procedures in a safe and eco-friendly way, leading to a full recovery (RF = 98–99%) of initial water permeation properties for membranes mainly containing Zr-MOF complexes. Nanoscale confinement of 2D materials was hence demonstrated to be proficient and suitable for controlling interfacial events that result in beneficial changes in the overall productivity and eco-friendliness of MD processes. The effectiveness of membranes was proven through MD testing under somewhat soft conditions. An increase of up to 17–19% was measured through membranes functionalized with aliquots of PNIPAMs/graphene and PNIPAMs/MOFs coming in contact with synthetic seawater (NaCl 35 gL^−1^). Flux through all engineered membranes was four to five times higher than in pristine PVDF membranes during the first hour of treatment with a mixture of sodium chloride (35 gL^−1^) and humic acid (1 mgmL^−1^), while better antifouling action was detected with increased time. This work demonstrates how the functionalization of PVDF membrane surfaces with aliquots of functional materials can be a more sustainable and fruitful route than incorporation of larger amounts of nanofiller into the bulk.

These novel responsive membranes can be regarded as the interplay of cooperative functions, which may well address the rationalization of new properties for functional PVDF membranes. This approach is aimed at paving the way for the realization of new families of active PVDF interfaces for more efficient, reliable, and sustainable recovery of reusable water by DCMD operations.

## Data Availability

Not applicable.

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
