# Peer review of "Aliquots of MIL-140 and Graphene in Smart PNIPAM Mixed Hydrogels: A Nanoenvironment for a More Eco-Friendly Treatment of NaCl and Humic Acid Mixtures by Membrane Distillation"

_membranes, 2023, doi:10.3390/membranes13040437_

Round 1

Reviewer 1 Report (Previous Reviewer 1)

Comments and Suggestions for Authors

The authors addressed all the comments. 

Reviewer 2 Report (Previous Reviewer 2)

Comments and Suggestions for Authors

I accept this version of manuscript

This manuscript is a resubmission of an earlier submission. The following is a list of the peer review reports and author responses from that submission.

Round 1

Reviewer 1 Report

Comments and Suggestions for Authors

Good topic to be investigated. The following need to be addressed: 

1- Better quality for SEM images.

2- Cross section of the membranes to better understand the depth of the surface modification. 

3- While the authors measured the contact angle for each sample, the Liquid Entry Pressure LEP is a critical value to be measured. 

4- More analysis should be provided for the pores. (what is the maximum pore, median pore size, and minimum pore size). what is the porosity percentage ( kindly check the experimental, the results, and discussion sections in this paper https://www.sciencedirect.com/science/article/abs/pii/S0011916422000583)

5- The work is not suitable for publications with the current quality of figures. Most of the Figures need to be improved (not clear, missing axis description,..etc). Captions for figures need to be corrected.

6- A figure with salt rejection over time needs to be presented.

6- Some important numbers need to be presented in the conclusion

Reviewer 2 Report

Comments and Suggestions for Authors

The review of Aliquots of MIL-140 and Graphene in Smart PNIPAM Mixed Hydrogels: A nanoenvironment for a more eco-friendly treatment of NaCl and humic acid mixtures by membrane distillation by Gugliuzza et al.

I find this paper as a good one. In generaln no critical remarks have been made by me.

I'm wondering about figure 1. Why is there a bimodal distribution here? This is a very unnatural situation. Maybe the authors would try to explain.

The quality of the figures is low. Almost all figures should be improved.
